# Biased belief updating and suboptimal choice in foraging decisions

Neil Garrett [1,2 ✉] & Nathaniel D. Daw [1]

Deciding which options to engage, and which to forego, requires developing accurate beliefs about the overall distribution of prospects. Here we adapt a classic prey selection task from foraging theory to examine how individuals keep track of an environment's reward rate and adjust choices in response to its fluctuations. Preference shifts were most pronounced when the environment improved compared to when it deteriorated. This is best explained by a trial-by-trial learning model in which participants estimate the reward rate with upward vs. downward changes controlled by separate learning rates. A failure to adjust expectations sufficiently when an environment becomes worse leads to suboptimal choices: options that are valuable given the environmental conditions are rejected in the false expectation that better options will materialize. These findings offer a previously unappreciated parallel in the serial choice setting of observations of asymmetric updating and resulting biased (often overoptimistic) estimates in other domains.

[1] Princeton Neuroscience Institute and Department of Psychology, Princeton University, Princeton, NJ 08544, USA. [2] Department of Experimental Psychology, University of Oxford, Woodstock Rd, Oxford OX2 6GG, UK. ✉email: ngarrett@princeton.edu

In contrast to classic economic decisions in which individuals choose from a menu of well-defined options presented simultaneously[1–7], the options in many real-world decisions are encountered serially and cannot directly be compared to one another. For instance, should I take a currently available option (e.g., hire this candidate for a position or accept this proposition of a romantic date)? Or should I forgo it in the expectation that a better option will come along? Insofar as accepting an option may require passing up unknown opportunities that might arise later, these choices involve opportunity cost and require comparing each prospect to some measure of the overall distribution of alternatives.

Such serial decisions arise canonically in several stylized tasks from animal ethology and optimal foraging theory, such as the prey selection task[8,9]. Here, a predator encounters a series of potential prey, which may each be accepted or forgone. If accepted, an option provides gain (e.g., calories) but pursuing and consuming it costs time that could instead be used to search for additional prey. The optimal choice rule, given by the Marginal Value Theorem (MVT)[10], is to accept an option only if its local return (calories divided by time) exceeds the opportunity cost of the time spent. This is just the calories per timestep that otherwise would be expected to be earned on average: the overall, long-run reward rate in the environment. Thus, foragers should be pickier when the environment is richer: a mediocre target (e.g., a skinny, agile animal that takes time to chase down) may be worthwhile in a barren environment, but not in one rich with better alternatives.

In this setting, choice turns largely on estimating the environment's present rate of return, which establishes the threshold against which each prospective prey item can be assessed. Importantly, the MVT specifies the optimal static choice policy in an environment, and thus predicts differences in behavior between environments. But it does not prescribe the dynamics of how an organism might attain the optimum. Accordingly, its predictions have most often been studied in terms of asymptotic behavior in stable environments. But estimating an environment's richness from experience with offers is in fact a learning problem, especially in dynamic environments in which the availability of resources fluctuates over time, as with the seasons[11–13].

Here we ask how humans update beliefs about their environment's rate of return in a prey selection task and extend a trial-by-trial learning model previously used for patch foraging[11,13] to account for deviations from the MVT that we observe. Participants are tasked with choosing whether to accept or reject serially presented options which vary in terms of points earned and time expended (if accepted). By manipulating the rate of reward between blocks, we are able to examine how individuals respond to changes in the overall richness of their environment as well as examine dynamic trial-by-trial adjustments that occur within blocks. Across three experiments, participants were sensitive to both global and trial-to-trial changes in the environment, in the direction predicted by the MVT[9,10]. But in all experiments, model estimates indicated that information integration was greater for positive compared to negative prediction errors. This asymmetry captures a key deviation we observed from the MVT-predicted policy: a reluctance to revise beliefs and change choices when environments deteriorate, leading in that circumstance to an overoptimistic bias and a pattern of overselective choices.

## Results

**Participants adapt to global fluctuations**. We first conducted two online experiments (see Fig. 1 and Methods for further details). On each trial, participants were offered an option (styled as an alien) and chose to accept or reject it. Accepting provided a reward (points, later converted to a bonus payment) but also

incurred an opportunity cost in the form of a time delay. There were four possible options (low delay, high reward: LDHR; low delay, low reward: LDLR; high delay, high reward: HDHR; high delay, low reward: HDLR; Fig. 1b). Following an accept decision, participants had to wait for the delay to elapse before the points were accrued and the next trial began. Following a reject decision, the experiment progressed to the next trial.

Participants were exposed to two blocks (environments, rich and poor), differing in the frequency of the best (LDHR) and worst (HDLR) options (Fig. 1c). In the rich environment, the best option outnumbered the others; in the poor environment, the worst option predominated. Block order was counterbalanced across participants, so that each either completed rich first (RichPoor) or the opposite (PoorRich). Note that the two intermediate options, LDLR and HDHR, were identical in profitability (i.e., reward per second) and occurred with identical frequency. To simplify the analysis, we collapse these options into one intermediate option category. Separating them does not change the pattern of results (see Supplementary Information Fig. 1). Experiments 1 and 2 differed only in the duration of each block (Experiment 1: 15 min per block; Experiment 2: 10 min per block).

The MVT predicts that acceptance should depend on an option's profitability and the overall quality of the environment. Accordingly, the decision whether to accept vs. reject an option was sensitive to both its profitability (i.e., reward per second) and also the environment (Fig. 2a, b). Specifically, a repeated measures ANOVA on the percentage of accept decisions with option (best, intermediate, worst) and environment (rich, poor) as repeated factors revealed a main effect of environment (Experiment 1: $F(1, 39) = 15.40$, $p < 0.001$, partial $\eta^2 = 0.28$; Experiment 2: $F(1, 37) = 8.38$, $p = 0.006$, partial $\eta^2 = 0.19$), a main effect of option (Experiment 1: $F(2, 78) = 540.41$, $p < 0.001$, partial $\eta^2 = 0.93$; Experiment 2: $F(2, 74) = 367.15$, $p < 0.001$, partial $\eta^2 = 0.91$) and an environment by option interaction (Experiment 1: $F(2, 78) = 70.27$, $p < 0.001$, partial $\eta^2 = 0.64$; Experiment 2: $F(2, 74) = 38.11$, $p < 0.001$, partial $\eta^2 = 0.51$). As predicted by the MVT, in each experiment the interaction was driven by a selective change between environments in acceptance of the intermediate options; this change was greater than the change in acceptance for the best or worst options (intermediate vs. best option, Experiment 1: $t(39) = 8.34$, $p < 0.001$, 95% CI [0.23, 0.38]; Experiment 2: $t(37) = 5.60$, $p < 0.001$, 95% CI [0.20, 0.42]; intermediate vs. worst, Experiment 1: $t(39) = 9.89$, $p < 0.001$, 95% CI [0.26, 0.40]; Experiment 2: $t(37) = 7.59$, $p < 0.001$, 95% CI [0.28, 0.48], two-tailed paired sample $t$-tests on the difference in acceptance rates between environments for intermediate versus either alternative). However, while this effect was in the direction predicted by the MVT (i.e., participants were more selective in the richer environment), the magnitude of the change was not as dramatic as theoretically predicted. This is due in part, as discussed below, to a slow adjustment in particular circumstances.

**Participants adapt to local fluctuations**. Since the quality of environments was not explicitly instructed, the foregoing results imply that subjects learn from experience how selective to be. Although the MVT itself does not prescribe dynamics, it does suggest a simple class of learning model[11,13]: dynamically estimate the reward rate in the environment (e.g., by an incremental running average), and then use this as an acceptance threshold in the MVT choice rule. Such learning predicts that choices should be sensitive not just to block-wise manipulation of the environmental richness, but also to local variation in offers, since for instance receiving a poor offer will incrementally decrease the

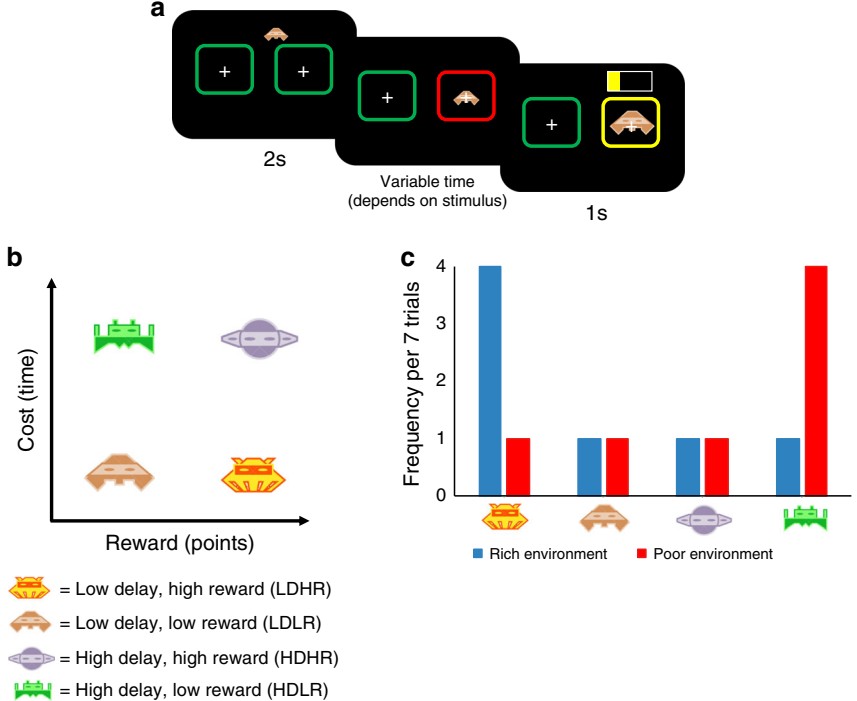

**Fig. 1 Behavioral task and variables. a** Timeline of the task. On each trial, an option (presented as one of four different aliens) approached one of two targets. Participants decided to accept the option by selecting the target the option approached (right target in this example) or reject the option by selecting the alternative target (i.e. the target the option did not approach, left target in this example). When options were accepted, the selected target changed color to red and participants were required to maintain a key press during which the option expanded within the target. The target then changed color to yellow indicating that participants could release the key press and the number of points obtained was displayed represented as a partially filled horizontal bar. When options were rejected, the experiment immediately progressed to the next trial. **b** On each trial, participants encountered one of four possible options each of which provided either a low/high reward (points, later converted to a bonus payment) but also incurred either a short/long opportunity cost in the form of a time delay. **c** The experiment was divided into two blocks (environments) and the frequency of the four different options varied in each of these. There was a rich environment in which the best option (low delay, high reward) outnumbered the other three options and a poor environment in which the worst option (high delay, low reward) outnumbered the other three options. The order of the blocks was counterbalanced between participants. Stimuli were designed by Sahua (https://www.123rf.com/profile_sahua) and are copyrighted property of 123RF Limited.

estimated reward rate, and incrementally decrease selectivity on the very next trial.

Accordingly, we examined evidence for such trial-to-trial learning by investigating whether the decision to accept an option fluctuated according to recent experience. We separated trials according to the option participants encountered on the previous trial, independent of the decision on the previous trial. To ensure any effect of local context was not merely driven by the block-wise environment type effect discussed above, we controlled for block type by entering environment (rich, poor) as a repeated factor in the analysis along with previous offer (best, inter-mediate, worst). This analysis revealed that participants increased their acceptance rates (over all options) on the current trial, the worse the option on the previous trial had been (main effect of previous option Experiment 1: $F_{(2, 78)} = 43.69$, partial $\eta^2 = 0.53$, $p < 0.001$; Experiment 2: $F_{(2, 74)} = 31.68$, $p < 0.001$, partial $\eta^2 = 0.46$, Fig. 2c, d). In other words, participants became more selective (less likely to accept the current option) immediately following evidence that the environment was rich (previous encounter with the best option) and less selective when it was poor (previous encounter with the worst option), consistent with an MVT-inspired learning model.

**Evidence of block-wise learning.** One additional piece of evidence speaking to the learning process was apparent in block order effects. In particular, we examined whether global fluctuations in acceptance rates were modulated by the order in which

the environments were encountered. We did this by implementing a new repeated measures ANOVA with option (best, intermediate, worst) and environment (rich, poor) as repeated factors, and this time included order condition (RichPoor, PoorRich) as a between-participant factor. This revealed an interaction between environment and order condition (Experiment 1: $F_{(1, 38)} = 12.11$, $p = 0.001$, partial $\eta^2 = 0.24$; Experiment 2: $F_{(1, 36)} = 7.24$, $p = 0.011$, partial $\eta^2 = 0.17$) as well as, as before, main effects of environment (Experiment 1: $F_{(1, 38)} = 18.22$, $p < 0.001$, partial $\eta^2 = 0.32$; Experiment 2: $F_{(2, 35)} = 8.00$, $p < 0.008$, partial $\eta^2 = 0.18$) and option (Experiment 1: $F_{(2, 76)} = 555.15$, $p < 0.001$, partial $\eta^2 = 0.94$; Experiment 2: $F_{(2, 72)} = 402.48$, $p < 0.001$, partial $\eta^2 = 0.92$).

The interaction reflected PoorRich participants (those who encountered the poor environment first) being more sensitive to the change in environments compared to RichPoor participants. That is, PoorRich participants (compared to the opposite order) showed a greater increase in their percentage of accept decisions in the poor environment compared to the rich environment (Supplementary Information Fig. 1). To better visualize this, we calculated a difference score for the change in acceptance rates (across all options, see Methods) between the poor environment and rich environment for each participant. Positive scores indicate an overall increase in acceptance rates in the poor environment relative to the rich environment. We then compared these difference scores for participants in the RichPoor condition versus the PoorRich condition (Fig. 3a, b). This revealed that

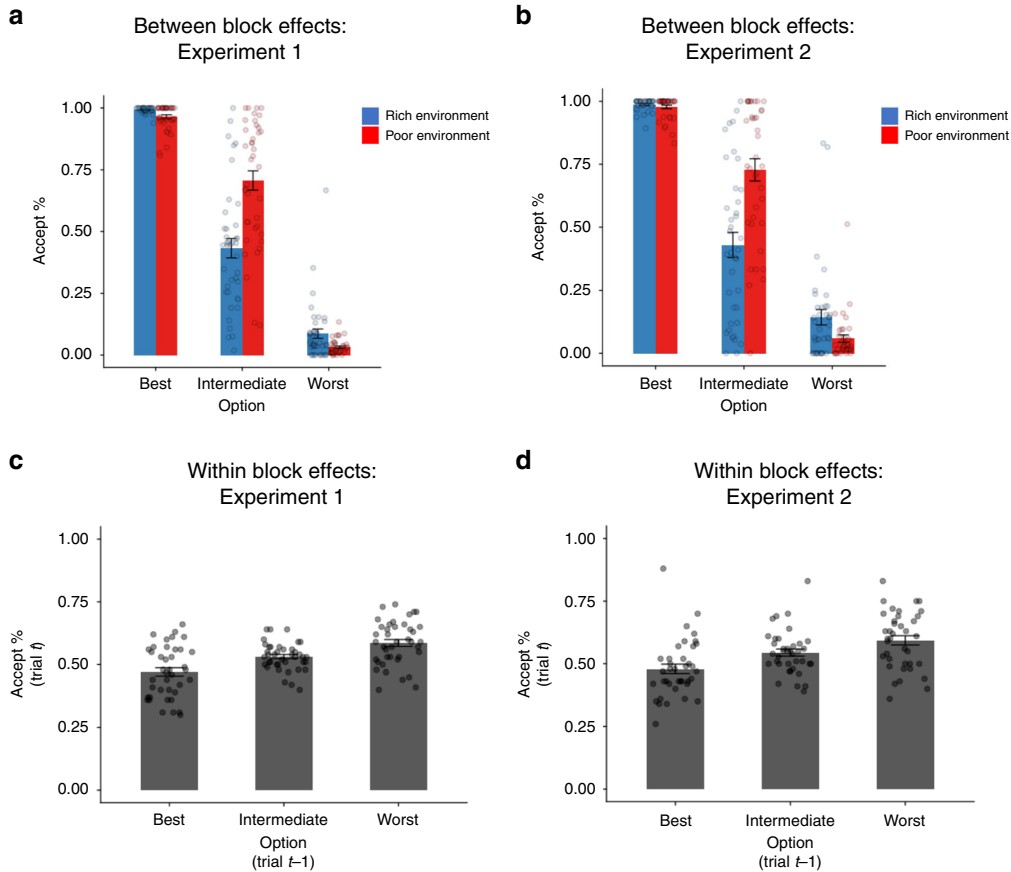

**Fig. 2 Global and local effects of the environment. a, b** Participants (Experiment 1, $N = 40$, Experiment 2, $N = 38$) increased their decision to accept according to how good each option was in terms of its reward per second and increased their decision to accept options overall in the poor compared to the rich environment. There was also an environment by option interaction driven by the change in acceptance between environments being greatest for the intermediate options compared to the best (Experiment 1: $t(39) = 8.34$, $p = 3.3094E-10$; Experiment 2: $t(37) = 5.60$, $p = 0.000002$, two-sided paired sample $t$-tests on the change in acceptance rates between environments for intermediate versus best option) and worst (Experiment 1: $t(39) = 9.89$, $p = 3.5216E-12$; Experiment 2: $t(37) = 7.59$, $p = 4.6782E-9$, two-sided paired sample $t$-tests on the change in acceptance rates between environments for intermediate versus worst option). **c, d** Acceptance rates were modulated by trial-to-trial dynamics. Participants (Experiment 1, $N = 40$, Experiment 2, $N = 38$) increased their acceptance rates for an option, the worse the previous option had been. Repeated measures ANOVA with previous option (best, intermediate, worst) and environment (rich, poor) as factors (main effect of previous option: Experiment 1: $F(2, 78) = 43.69$, $p = 1.864E-13$; Experiment 2: $F(2, 74) = 31.68$, $p = 1.1497E-10$). Best = high reward, low delay option; intermediate = low reward, low delay and high reward, high delay options combined; worst = low reward, high delay option. Dots represent individual data points; bars represent the group mean. Error bars represent mean ± standard error of the mean. Source data are provided as a Source Data file.

PoorRich participants showed a significant increase in their percentage of accept decisions in the poor environment compared to the rich environment (Experiment 1: $t(20) = 6.79$, $p < 0.001$, 95% CI [0.122, 0.230]; Experiment 2: $t(20) = 4.99$, $p < 0.001$, 95% CI [0.104, 0.254], two-tailed one-sample $t$-test on the difference scores versus 0) in contrast to RichPoor participants who exhibited only a marginally higher rate of acceptance in the poor compared to the rich environment in Experiment 1 ($t(18) = 1.97$, $p = 0.07$, 95% CI [−0.0035, 0.1048]) which was not significant in Experiment 2 ($t(16) = 1.17$, $p = 0.26$, 95% CI [−0.046, 0.159]) with there being a significant difference between these two groups in the difference scores (Experiment 1: $t(38) = 3.41$, $p = 0.002$, 95% CI [0.051, 0.199]; Experiment 2: $t(36) = 2.08$, $p = 0.045$, 95% CI [0.003, 0.242], two-tailed independent sample $t$-tests, comparing difference scores for PoorRich versus RichPoor participants).

**Computational modeling.** We reasoned that this asymmetry reflected the operation of the underlying learning rule by which subjects adjusted their behavior from the poor to the rich

environment or vice versa. Accordingly, we fit the behavioral data to two reinforcement learning models: a Symmetric Model and an Asymmetric Model. In each of these models, participants were assumed to maintain an ongoing estimate of the reward rate (reward per second), $\rho$, which updated every second. This estimate was then used on each trial to calculate the opportunity cost of accepting an option ($\rho$ multiplied by the option's time delay) at the time of choice. Under this model specification, even though the absolute cost (in terms of number of seconds delay) imposed by accepting a specific option was the same throughout the task, the opportunity cost could vary according to participant's current estimate of $\rho$. The decision to accept or reject was modeled as a comparison between the opportunity cost of accepting an option (effectively the value of rejecting) against the reward that the option would collect (effectively the value of accepting). This was implemented using a softmax decision rule with an inverse temperature parameter ($\beta_1$) governing the sensitivity of choices to the difference between these two quantities, and an intercept ($\beta_0$) capturing any fixed, overall bias toward or against acceptance (see Methods for further details on the model fitting procedure). Both

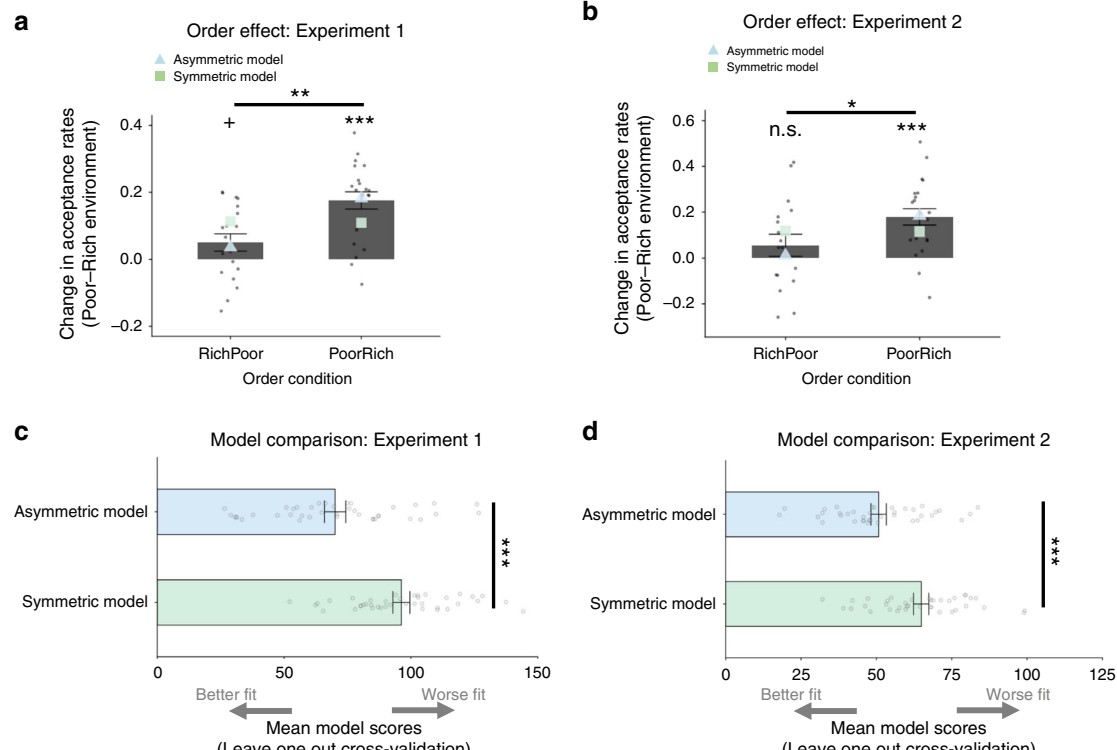

**Fig. 3 Order effect and model comparison. a** Participants who experienced the poor environment followed by the rich environment (PoorRich group, $N =$ 21) changed choices between environments to a greater extent than participants who experienced a rich environment followed by a poor (RichPoor group, N = 19) environment ($t(38) = 3.41$, $p = 0.002$, two-tailed independent $t$-test comparing the change in acceptance rates between these two groups of participants). The Asymmetric Model was able to recapitulate this order effect by having separate learning rate for when reward rate estimates increased and when they decreased. The Symmetric Model which had a single learning rate predicted that the change in acceptance rates between environment ought to be the same for RichPoor and PoorRich participants. Gray dots represent individual data points and gray bars represent the group mean. Blue triangles represent the pattern of choices generated by simulations from the Asymmetric Model. Green squares represent the pattern of choices generated by simulations from the Symmetric Model. **b** This pattern of results replicated in a second experiment ($t(36) = 2.08$, $p = 0.045$, two-tailed independent $t$-test comparing the change in acceptance rates between these two groups of participants) (PoorRich group, $N = 21$; RichPoor group, $N = 17$). **c** The Asymmetric Model provided a superior fit to the data than the Symmetric Model in Experiment 1 ($t(39) = 7.20$, $p = 1.1551E-8$, two-tailed paired sample $t$-tests comparing LOOcv scores for the Asymmetric versus the Symmetric Model. Plotted are the mean LOOcv scores over subs ($N = 40$) for each model. **d** The Asymmetric Model provided a superior fit to the data than the Symmetric Model in Experiment 2 as well ($t(37) = 6.04$, $p = 5.4824E-7$, two-tailed paired sample $t$-tests comparing LOOcv scores for the Asymmetric versus the Symmetric Model). Plotted are the mean LOOcv scores over subs ($N = 38$) for each model. Dots represent individual data points and bars represent the group mean. Error bars represent mean ± standard error of the mean. $^{+}0.05 < p < 0.10$; $^{*}p < 0.05$; $^{**}p < 0.01$; $^{***}p < 0.001$: independent sample $t$-test/paired sample $t$-test/one-sample $t$-test (vs. 0) as appropriate (all two tailed) n.s. non-significant. Source data are provided as a Source Data file.

models used a delta-rule running average[14] to update $\rho$ according to positive and negative prediction errors. Negative prediction errors were generated every second that elapsed without a reward (for example, each second of a time delay). Positive prediction errors were generated on seconds immediately following a time delay when rewards were received. Note therefore that updates to the reward rate are dependent on the decision of the participant. This captures a key feature of the prey selection problem[9]: that the relevant decision variable is the participant's average earnings given their status quo choice policy, so an option merely offered (but not accepted) should not count.

The difference between the Symmetric Model and the Asymmetric Model was whether there were one or two learning rate parameters. The Symmetric Model contained just a single learning parameter, $\alpha$. This meant that $\rho$ updated at the same rate regardless of whether the update was in a positive or a negative direction. The Asymmetric Model had two learning parameters: $\alpha^{+}$ and $\alpha^{-}$. This enabled $\rho$ to update at a different rate, according to whether the update was in a positive ($\alpha^{+}$) or a negative ($\alpha^{-}$) direction. This causes an overall (asymptotic) bias in $\rho$ (e.g., if $\alpha^{+} > \alpha^{-}$, then the environmental quality is overestimated), but

also dynamic, path-dependent effects due to slower adjustment to one direction of change over the other.

**Asymmetric Model a better fit to choice data.** We fit the per-participant, per-trial choice timeseries to each model, and compared models by computing unbiased per-participant negative log marginal likelihoods via subject-level Leave One Out cross-validation (LOOcv) scores for each participant. The Asymmetric Model provided a superior fit to the choice data than the Symmetric Model both in Experiment 1 ($t(39) = 7.20$, $p < 0.001$, 95% CI [18.77, 33.45], two-tailed paired sample $t$-tests comparing LOOcv scores for the Asymmetric versus the Symmetric Model, Fig. 3c and Table 1) and in Experiment 2 ($t(37) = 6.04$, $p < 0.001$, 95% CI [9.39, 18.86], Fig. 3d and Table 1). At the population level, formally comparing the two learning rates (see Methods) in the Asymmetric Model revealed that this asymmetry was, on average, significantly biased toward $\alpha^{+} > \alpha^{-}$ (Experiment 1: $z = 2.83$, $p < 0.01$; Experiment 2: $z = 3.73$, $p < 0.001$, Table 1). This meant that prediction errors that caused $\rho$ to shift upwards (following receipt of a reward) had a greater impact than prediction errors that caused $\rho$ to shift downwards (following the

**Table 1 Model fitting and parameters across the three experiments.**

| Experiment/Model | LOOcv | $\alpha$ | $\alpha^+$ | $\alpha^-$ | $\beta_0$ | $\beta_1$ |
|---|---|---|---|---|---|---|
| *Experiment 1 (N = 40)* | | | | | | |
| Symmetric Model | 96.22 (3.34) | 0.0218 [95% CI = 0.0057, 0.0657] | - | - | 0.9855 [95% CI = 0.6755, 1.2954] | 0.076 [95% CI = 0.0701, 0.0819] |
| Asymmetric Model | 70.10*** (4.19) | - | 0.005 [95% CI = 0.0027, 0.0087] | 0.0016 [95% CI = 0.0008, 0.0031] | −1.8179 [95% CI = −2.571, −1.064] | 0.0748 [95% CI = 0.0660, 0.0837] |
| *Experiment 2 (N = 38)* | | | | | | |
| Symmetric Model | 64.89 (2.54) | 0.0278 [95% CI = 0.0085, 0.0749] | - | - | 0.846 [95% CI = 0.5742, 1.1178] | 0.0747 [95% CI = 0.0685, 0.0810] |
| Asymmetric Model | 50.77*** (2.56) | - | 0.0047 [95% CI = 0.0031, 0.0070] | 0.0018 [95% CI = 0.0011, 0.0027] | −1.4624 [95% CI = −2.1052, −0.8197] | 0.0742 [95% CI = 0.0659, 0.08250] |
| *Experiment 3 (N = 38)* | | | | | | |
| Symmetric Model | 80.44 (3.82) | 0.123 [95% CI = 0.0528, 0.2403] | - | - | 0.781 [95% CI = 0.4934, 1.0691] | 0.0751 [95% CI = 0.0655, 0.0848] |
| Asymmetric Model | 70.29*** (3.76) | - | 0.0279 [95% CI = 0.0187, 0.0404] | 0.0152 [95% CI = 0.0104, 0.0219] | −1.088 [95% CI = −1.6028, −0.5728] | 0.07 [95% CI = 0.0583, 0.0819] |

The table summarizes for each model its fitting performances and its average parameters: LOOcv: Leave One Out cross-validation scores, mean (standard error of the mean) over participants, lower LOOcv scores indicate better performance; $\alpha$: learning rate for both positive and negative prediction errors (Symmetric Model); $\alpha^+$: learning rate for positive prediction errors (Asymmetric Model); $\alpha^-$: average learning rate for negative prediction errors (Asymmetric Model); $\beta_0$: softmax intercept (bias towards reject); $\beta_1$: softmax slope (sensitivity to the difference in the value of rejecting versus the value of accepting an option). Data for model parameters are expressed as mean and 95% confidence intervals (calculated as the sample mean ± 1.96 × standard error).
***$P < 0.001$ comparing LOOcv scores between the two models, two tailed paired sample $t$-test (Experiment 1: $t(39) = 7.20$, $p = 1.1551\text{E}−8$; Experiment 2: $t(37) = 6.04$, $p = 5.4824\text{E}−7$; Experiment 3: $t(37) = 5.47$, $p = 0.000003$).
Source data are provided as a Source Data file.

absence of a reward). Individually, 88% of participants in Experiment 1 and 92% of participants in Experiment 2 were estimated to have a higher learning parameter for positive ($\alpha^+$) compared to negative ($\alpha^-$) errors. To examine whether asymmetric learning was a stable feature of learning, or rather emerged only after the environment switch (e.g., in response to expectations set up by the first block), we also reran both models, this time fitting only the data from the first environment participants encountered. The Asymmetric Model was again a better fit to the data than the Symmetric Model (Experiment 1: $t(39) = 5.56$, $p < 0.001$, 95% CI [4.83, 10.36]; Experiment 2: $t(37) = 6.01$, $p < 0.001$, 95% CI [3.90, 7.87], two-tailed paired sample $t$-test) with $\alpha^+ > \alpha^-$ in the Asymmetric Model (Experiment 1: $z = 5.64$, $p < 0.01$; Experiment 2: $z = 9.96$, $p < 0.01$).

**Asymmetric Model accounts for order effect.** Next, we examined the impact of this asymmetry when the environment changed. Importantly, the model carried $\rho$ over between environments rather than resetting at the start of a new block. (This feature was motivated by the block order effect, which rejects the otherwise identical model that resets values to some constant at each block and thus predicts order invariance[15].) Accordingly, participants had to update beliefs that had been established in the first environment they encountered (poor for the PoorRich group, rich for the RichPoor group). Given that the learning asymmetry was revealed to be in a positive direction ($\alpha^+ > \alpha^-$) we reasoned that this update ought to occur faster when going from a poor environment into a rich environment, as large rewards become more commonplace (i.e. the PoorRich group), which would explain the block order effect.

Indeed, simulating the experiment using a population of subjects drawn according to the best-fitting parameter distributions for each model (Fig. 3a, b), we found that both models reproduced an overall effect of environment type, but only the Asymmetric Model captured the block order effect. Returning to the fits to actual participants' choices, we further unpacked the model's account of this effect by extracting trial-by-trial estimates of $\rho$ from the model's fit to each trial and participant and entering them into a repeated measures ANOVA with environment (rich, poor) as a repeated factor and order condition (RichPoor, PoorRich) as a between-participant factor. This revealed a pattern in accord with the intuition that the global effect arose from

slower adjustment of the acceptance threshold by the RichPoor group. In particular, there was a significant environment by condition interaction (Experiment 1: $F(1, 38) = 14.67$, $p < 0.001$, partial $\eta^2 = 0.28$, Fig. 4a; Experiment 2: $F(1, 36) = 21.42$, $p < 0.001$, partial $\eta^2 = 0.37$, Supplementary Information Fig. 2). This arose out of a significant difference in $\rho$ between environments for participants in the PoorRich condition (Experiment 1: $t(20) = 8.64$, $p < 0.001$, 95% CI [5.98, 9.79]; Experiment 2: $t(20) = 6.08$, $p < 0.001$, 95% CI [4.56, 9.32], two-sided paired sample $t$-test) which was absent among RichPoor participants (Experiment 1: $t(18) = 1.16$, $p = 0.26$, 95% CI [−1.32, 4.54]; Experiment 2: $t(16) = 0.004$, $p = 0.997$, 95% CI [−1.87, 1.88]).

**Greater exposure to poor environment reduces order effect.** To further probe whether asymmetric learning caused strategies to perseverate among RichPoor participants, we ran a third cohort of participants. The procedure was exactly as for Experiments 1 and 2 but with one key difference. Now, between the two environments (either between the Rich and the Poor environment or between the Poor and the Rich environment) we inserted a third environment in which participants only saw the worst option (HDLR, see Methods for full details). We predicted that with the addition of this environment, participants in the RichPoor condition would now have sufficient exposure to a lean environment to allow their expectations to adjust before entering the poor environment. In doing so, we predicted that we ought to no longer to observe a difference in the change in acceptance rates between environments for RichPoor and PoorRich participants.

The Asymmetric Model once again proved a better fit to choice behavior compared to Symmetric Model ($t(37) = 5.47$, $p < 0.001$, 95% CI [6.39, 13.91], two-tailed paired sample $t$-tests comparing LOOcv scores for the Asymmetric Model versus the Symmetric Model, see also Table 1) with $\alpha^+$ being greater than $\alpha^-$ ($z = 2.28$, $p < 0.05$). But in contrast to before, both groups significantly changed their acceptance rates between environments (RichPoor: $t(14) = 3.07$, $p = 0.008$, 95% CI [0.04, 0.21]; PoorRich: $t(22) = 4.47$, $p < 0.001$, 95% CI [0.09, 0.24], two-tailed one-sample $t$-test versus 0) and there was no difference in this change in acceptance rates between RichPoor and PoorRich participants ($t(36) = 0.77$, $p = 0.45$, 95% CI [0.07, 0.15], two-tailed independent sample $t$-test, Fig. 4b).

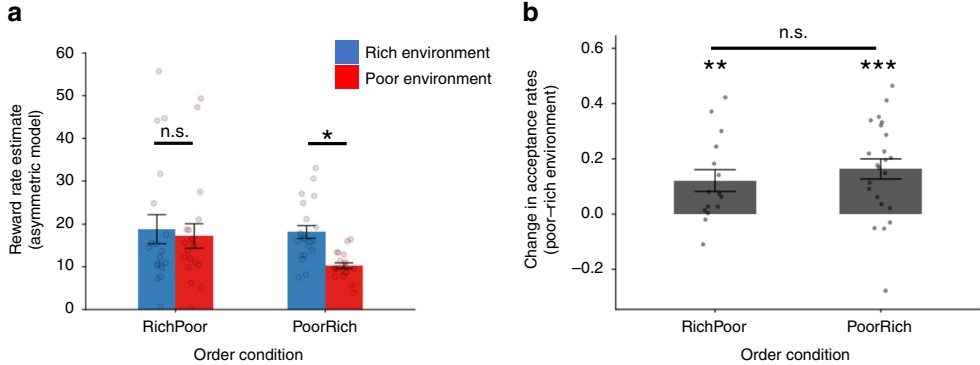

**Fig. 4 Reward rate change and Experiment 3. a** Extracting reward rate estimates ($\rho$) from the Asymmetric model for each participant in each experimental block in Experiment 1 (PoorRich group, $N = 21$; RichPoor group, $N = 19$) revealed significant environment by condition interaction ($F(1, 38) = 14.67$, $p = 0.000465$). This arose out of a significant difference in $\rho$ between environments for participants in the PoorRich condition ($t(20) = 8.64$, $p = 3.4864E{-}8$, two-tailed paired sample $t$-test) which was absent among participants assigned to the RichPoor condition ($t(18) = 1.16$, $p = 0.26$, two-tailed paired sample $t$-test). **b** In Experiment 3, both groups of participants (PoorRich group, $N = 23$; RichPoor group, $N = 15$) significantly changed their acceptance rates between environments (RichPoor: $t(14) = 3.07$, $p = 0.008$; PoorRich: $t(22) = 4.47$, $p = 0.000193$, two-tailed one-sample $t$-test vs 0) and there was no longer any difference in the change in acceptance rates between RichPoor and PoorRich participants ($t(36) = 0.77$, $p = 0.45$, two-tailed independent sample $t$-test). \*$p < 0.05$, paired sample $t$-test; n.s. non-significant ($p > 0.05$, two sided); \*\*$p < 0.01$, one sample $t$-test (vs 0, two tailed); \*\*\*$p < 0.001$, one sample $t$-test (vs 0, two tailed). Dots represent individual data points and bars represent the group mean. Error bars represent mean ± standard error of the mean. Source data are provided as a Source Data file.

## Discussion

Across three experiments using a prey selection task, we find that individuals adjusted their choices both between environments—becoming less selective in a poor compared to a rich environment—and within environments—altering selectivity on a trial-by trial basis according to the previous encounter. The block-wise effect is consistent with foraging theory[9] and previous empirical data both from animals[9,10,16–19] and humans[12,20–23]. The trial-wise effect is a rather direct prediction of incremental learning rules for the acceptance threshold, which have previously been proposed and studied in terms of a different class of foraging tasks, patch leaving tasks[11,24–26], where choice-by-choice effects are harder to observe due to the task structure.

However, inconsistent with optimal foraging theory, we observed interesting suboptimalities in individuals' choices. In particular, adjustments in choices between environments were attenuated when an individual's environment became worse compared to when the environment improved. The experience-dependent nature of this bias strongly suggests that it is rooted in learning, and we present modeling showing that it can be understood in terms of an asymmetric learning rule, which scales positive and negative prediction errors differently. Such decomposition of learning by valence is a recurring theme in decision neuroscience, but its consequences in the foraging setting—path-dependent biases in estimates of opportunity cost, leading to systematic choice biases—have not previously been appreciated. Indeed, optimism biases of the sort implied by these models may have particularly important effects in many real-world foraging-like tasks (including hiring and employment decisions and mate selection) because these learned estimates play such a key role in choice: encountering opportunities serially forces the decision-maker to compare them to a learned (and potentially biased) estimate of the other fish in the sea, rather than directly to their alternatives.

We were able to account for the deviations from foraging theory by augmenting a learning rule that had previously been used in the patch foraging setting[11,13]. Across all three experiments, endowing the model with separate learning rates for positive and negative adjustments to the environment's reward rate provided a better qualitative and quantitative fit to participants' choices compared to a baseline model that did not

distinguish between these two types of adjustments. This feature provided the model with the capacity to shift estimates of the environments reward rate up and down at different rates, enabling it to recapitulate the differences in choice adjustments depending on the sequence in which participants experienced rich and poor environments.

Interestingly, the order effect we observe, and the learning account of it, indicate that participants carry over information about the reward rate from one environment into the next. This is despite the fact that participants are explicitly told at the start of the experiment that they will experience different environments in the task and, during the experiment, new environments are clearly signaled. It may be that a variety of factors such as the introduction of more transitions, more environments, time spent in each environment, volatility, and/or greater contextual differences between environments might prompt a shift towards resetting or reinstating previously learned reward rates thereby mitigating the carryover biases we observe.

Our learning-based account is further supported, at least to a limited extent, by our third experiment, in which the block order effect was not significant in a version of the task that included additional experience with a poor reward rate. This negative finding is predicted by the learning model, and tends to mitigate against other explanations (e.g., ones in which the block order asymmetry relates to some sort of primacy or anchoring on the initial experience) that would predict equivalent effects to the (similarly designed and powered) Experiments 1 and 2. Of course, null effects (even predicted ones) should be interpreted with caution, and the difference in significance between experiments does not necessarily imply the effects are themselves different. Future experiments including both conditions in a single design will be required to permit their direct statistical comparison.

One reason that learning of a global environmental reward estimate, as studied here, is important is that this variable plays quite a ubiquitous role governing many aspects of choice. Apart from serving as the decision variable for prey foraging and patch leaving, the same variable also arises in theoretical and experimental work on physical vigor[27] and cognitive effort[28], deliberation[29], action chunking[30], risk sensitivity and time discounting[31], mood and stress[25]. Furthermore, whereas in optimal foraging theory[9] and many of these other examples, the

average reward is thought to play a direct role in choice as a decision variable, a similar global average reward estimating can also affect choice more indirectly, by serving as a comparator during learning so as to relativize what is learned to a contextual reference[32,33]. In this respect, our study parallels another recent line of work[34–36] that studies choices between options originally trained in different contexts to reveal effects of a global contextual average reward on relativizing learned bandit valuations. It seems likely that these learning-mediated effects are a different window on the same underlying average reward estimates driving choice in the current study, and (for instance) vigor in others[37]. It has not been tested whether hallmarks of learning asymmetry as we report here can also be detected in these other paradigms.

There is also another respect in which these results may reflect a broader feature of learning. In a variety of other domains, beliefs are more readily updated when people receive desirable compared to undesirable information. This pattern emerges when people receive information about the likelihood of different life events occurring in the future[38–41], information about their financial prospects[42], feedback about their intellectual abilities[43,44], personality[45], and physical traits such as attractiveness[43]. Although the results we present are consistent with these past instances of asymmetric learning, we believe it of note to find that it extends to a foraging setting. This setting captures a key aspect of many of the real-world choices humans undertake whereby an exhaustive explicit menu of all options available does not exist. As a result, choices are heavily reliant on subjective beliefs about what options will materialize in the future. Optimistic or pessimistic biases in this aspiration level, should they occur, can cause costly errors due to excessive rejection of decent offers, or, respectively, over-acceptance of objectively inadequate ones. By comparison, although asymmetric learning has also been observed in bandit tasks where subjects choose between a set of options, its effects on earnings are less pernicious in this setting because it biases evaluations of all the options being compared in the same direction and tends therefore to wash out. Thus, overall, the finding of asymmetric biases in foraging is a mechanism by which a wide range of suboptimal decisions could potentially arise.

Neural accounts of learning have also stressed the separation of negative and positive information and updates, which (given that firing rates cannot be negative) are potentially represented in opponent systems or pathways[3,33,46]. Appetitive and aversive expectancies, and approach and avoidance, are also associated with engagement of partly or altogether distinct brain regions[34,47–51]. More particularly, the direct and indirect pathways through the basal ganglia have been argued to support distinct pathways for action selection versus avoidance[3,46], with positive and negative errors driving updates toward either channel. These types of models also typically incorporate asymmetric updating, though not necessarily consistently biased in all circumstances toward positive information ($\alpha^+ > \alpha^-$).

One important interpretational caveat, at least with respect to the apparent analogy with other work on biased updating is that, compared to these other cases, in the foraging scenario we use here, upward and downward adjustments in reward rate estimates differ in ways other than valence. Specifically, upward errors are driven by reward receipt and downward ones by the passage of time, events which may be perceived and processed differently in ways that may also contribute to asymmetric learning. Future experiments will be required to uncover if valence—whether a piece of information is good or bad—is the key attribute that gives rise to the asymmetry we observe.

Another question that awaits future work is whether the learning-based mechanism we describe also contributes to biases that have been observed in other foraging scenarios. Notably, a number of other reported biases, especially in patch foraging

tasks, tend to reflect over-acceptance or overstaying[11,25,52]. Asymmetric learning of the sort modeled here offers a class of explanations for this longstanding and puzzling pattern of deviations from the predictions of optimal foraging theory. However, if overstaying is to be explained by asymmetric learning, it would imply pessimistic rather than the optimistically biased opportunity costs we observe here, and a bias toward negative updating rather than positive updating.

This apparent contrast between over-selectivity in the current study, compared to over-acceptance in others, in turn, raises a further experimental and theoretical question: what features of experience or environment determine the balance in sensitivity to positive and negative prediction errors[34]? Recent theoretical work has proposed that positively ($\alpha^+ > \alpha^-$) and negatively ($\alpha^- > \alpha^+$) biased beliefs can each be advantageous, depending on circumstances[53,54]. For example, computational simulations have suggested that overconfidence about one's own abilities can be beneficial in relatively safe environments (where the potential cost of error is lower than the potential gains). But such overconfidence is detrimental in threatening environments in which costs exceed gains[53]. In two arm bandit tasks, it has also been shown that a positive learning asymmetry can be advantageous over the long run if payouts from each bandit are rare. But conversely a negative learning asymmetry is advantageous if payouts are common[54]. However, the logic of all of these predictions is specific to the modeled class of tasks, and does not appear to extend straightforwardly to the foraging setting. Thus, the current results underline the need for future theoretical work clarifying to what type of foraging situation such a bias might be adaptive or adapted. Relatedly, the demonstration (in other tasks) that different circumstances favor different asymmetries has led to the prediction that people should adjust their bias to circumstance via some type of meta-learning[54,55]. To date though, empirical work has failed to find evidence for such an adaptive adjustment in bandit tasks[56,57], although there is evidence that biases in updating beliefs about oneself do flexibly adjust under threat[38]. Here, too, there remains the need for both theoretical and experimental work investigating how such adjustment plays out in the foraging scenario.

The possibility of biased reward rate evaluations, and particularly that they might be pessimistic in some circumstances, may also be of clinical relevance. Huys et al.[58] propose that a pessimistic estimate of an environment's rate of reward may offer a common explanation for a range of diverse symptoms of Major Depressive Disorder (MDD) such as anergia, excessive sleeping, lack of appetite, and psychomotor retardation. There is empirical evidence for learning rates becoming less positively biased in patients with MDD in the case of some (non-foraging related) tasks[59,60] but not in the case of others[61]. However, to date no empirical work has examined learning asymmetries among MDD patients in foraging related tasks, where the content of the beliefs being updated (the reward rate of an environment) could underlie these different MDD symptoms if pessimistic. Whether the learning asymmetry we observe disappears or reverses toward a negative direction in MDD patients is therefore an important question for future research.

## Methods

**Participants.** A total of 62 participants were recruited for Experiment 1. Twenty-two of these were excluded leaving a final sample of 40 participants (mean [standard deviation] age: 32.73 [7.94]; 11 female). A total of 55 participants were recruited for experiment 2. Seventeen of these were excluded leaving a final sample of 38 participants (mean [standard deviation] age: 33.71 [8.83]; 16 female; 2 gender undisclosed). Participants were recruited online via Amazon Mechanical Turk. Following best practice for studies with this population[62], several a priori exclusion criteria were applied to ensure data quality.

Participants were excluded if any of the following applied: (1) did not finish the task ($n = 5$); (2) made 20 or more missed responses ($n = 14$); (3) made 10 or more incorrect force trial responses (either accepting options when forced to reject or rejecting options when forced to accept, $n = 1$); (4) poor discriminability, defined as choosing the worst option (low reward, high delay) a greater percentage of times than the best option (high reward, low delay) or accepting/rejecting all options on every single trial ($n = 2$). Participants were paid $1.5 plus a bonus payment between $1.5 and $4 depending on performance in the task. All participants provided fully informed consent. The study design complied with all relevant ethical regulations and was approved by Princeton University's Institutional Review Board.

**Behavioral task**. The task began by asking participants to provide basic demographic information (age and gender) (Fig. 1). After providing this information, participants read task instructions on screen at their own pace and undertook a practice session for approximately 4 min. The two options and the environment used in the training session were not used in the actual task. After completing the training session, participants were then required to attempt a multiple-choice quiz to check their understanding of the task and instructions. Participants needed to get all the questions correct to pass the quiz and continue to the task. Participants who failed the quiz were required to reread the instructions and attempt the quiz again. (They were not asked to retake the training again.)

The main experiment comprised two different environments, each lasting 15 (Experiment 1) or 10 (Experiment 2) minutes in a block design (i.e., participants completed 15/10 min of one environment and then switched to the other environment for a new period of 15/10 min). Participants were told in the instructions that they would encounter the same options in each environment but the frequency of the options could vary between the environments and that their goal was to gain as collect as many points as possible in the time available. There was a break in-between the two environments. The background color was different for each environment and before continuing to the second environment; participants were explicitly told that they would now experience a new environment.

On each trial, one of four stimuli (options) appeared and moved toward one of two targets: one on the left side of the screen and another on the right. Participants accepted an option by selecting the target the option approached or rejected an option by selecting the alternate target. Participants had 2 s to respond by selecting the left/right target using separate keys on their computer keyboard. Following acceptance of an option, participants faced a time delay (1 or 7 s depending on the stimulus) during which they were required to keep the key used to accept the option pressed down. During this time the option on screen gradually approached them. At the end of the time delay, the target turned yellow and the number of points obtained appeared above the target (1 s) at which point participants could release the key press. Following rejection of an option, the experiment progressed to the next trial. If the participant failed to respond during the encounter screen or released a keypress before a time delay had finished, they faced a timeout of 8 s. This meant that having made the decision to accept an option, it was disadvantageous to then abort. Note that the delays and reward associated with each option were not previously learnt or instructed to participants. Participants sequentially learnt these during the task itself following an accept decision when they then observed the time required to capture the accepted option and, following this, observed the reward collected.

As an attention check and to encourage learning, 25% of trials were forced choice trials. On these trials, participants saw a red asterisk (*) appear over one of the two targets and had to choose that target. This meant that if it appeared over a target that a stimulus was approaching, they had to accept the stimulus on that trial. If it appeared over a target the stimulus was not approaching, they had to choose to reject the stimulus on that trial. Participants were told that more than five incorrect forced choice trials would see their bonus payment reduced by half. At the end of the experiment, participants were told how many points they had amassed in total and their corresponding bonus payment.

The task was programmed in JavaScript using the toolbox jsPsych[63] version 5.0.3.

**Stimuli**. Four different stimuli provided participants with one of two levels of points (presented to participants as 20/80% the length of a horizontal bar displayed on the reward screen (Fig. 1a) which corresponded to 20/80 points) and incurred one of two time delays (2 or 8 s including the 1 s for reward screen display). These options therefore assumed a natural ordering in terms of their value from best (low delay high reward (LDHR)), intermediate (low delay low reward (LDLR), high delay high reward (HDHR)) to worst (high delay low reward (HDLR)). The frequency of each option varied between the two different environments. In the rich environment, the best option was encountered four times more than the other three options. In a sequence of seven trials, the participant would encounter (in a random order) LDHR four times, and also encountering each of LDLR, HDHR and HDLR once. In the poor environment, in a sequence of seven trials, the participant would encounter HDLR four times and the other three options (LDHR, LDLR and HDHR) once (see Fig. 1c).

**Behavioral analysis**. To examine changes in acceptance rates between environments, we calculated the percentage of accept decisions for each participant for each option they saw in each environment. Forced choice trials and missed responses were excluded from this analysis. They are not, however, excluded from the computational models. To simplify the analysis presented in the main paper we collapsed the two intermediate options—which have an identical profitability (i.e. reward per second) of 10 points per second—together. We then entered these percentages into a repeated measures ANOVA with option (best/intermediate/worst) and environment (rich/poor) as repeated factors. We also ran the same ANOVA treating the two intermediate options as separate levels. In this instance, option (LDHR/LDLR/HDHR/HDLR) and environment (rich/poor) were entered as repeated factors (Supplementary Information).

To examine how changes in acceptance rates related to the order with which participants encountered the environment, we entered the percentage of accept decisions into a separate repeated measures ANOVA with option (LDHR/intermediate/HDLR) and environment (rich/poor) as repeated factors. Condition (RichPoor or PoorRich), which indicates which ordering of environments participants faced, was entered as a between-participant factor. As above, we also ran the same ANOVA treating the two intermediate options as separate levels (Supplementary Information Fig. 1).

To better characterize order effects, we calculated the difference in acceptance rates between environments (poor minus rich) for each option (LDHR, LDLR, HDHR, and HDLR). Hence positive scores indicate an increase in acceptance rates in the poor environment compared to the rich environment. We then calculated the mean change across the four options as a measure of overall change in acceptance rate between environments. To calculate whether this change was significant for each group separately, we conducted a one sample $t$-test (versus 0, two tailed). We compared the overall change in acceptance scores between participants (RichPoor and PoorRich) using independent sample $t$-tests (two tailed).

Finally, to examine trial-to-trial fluctuations, we first partitioned trials according to the option presented (best, intermediate, worst). We then calculated separately for each participant and for each environment, the percentage of times on the next trial the decision was to accept. We entered these acceptance scores into a $3 \times 2$-way repeated measure ANOVA with option (best, intermediate, worst) and environment (rich, poor) as factors. To visualize these fluctuations (Fig. 2c, d), we calculated the average of the two acceptance scores—one score for each environment—for each option.

**Computational models**. The optimal policy from the MVT is to accept an option (indexed by $i$) whenever the reward ($r_i$) that the option obtains exceeds the opportunity cost ($c_i$), of the time taken to pursue the option. This opportunity cost ($c_i$) is calculated as the time ($t_i$) that the option takes to pursue (in seconds) multiplied by the estimated reward rate (per second) of the environment at the current point in time ($\rho_t$):

$$c_i = \rho_t t_i. \tag{1}$$

Participants should therefore accept an option whenever $r_i \geq c_i$

Note we assumed that the quantities $r_i$ and $t_i$ were known to participants from the outset since they were easily observable and each of the four options ($i = \{1, 2, 3, 4\}$) always provided the exact same $r_i$ and $t_i$. But models that dropped this assumption (and instead assumed $r_i$ and $t_i$ were learned via experience rather than known) provided similar patterns of results (see Supplementary Information Learn Options Models).

We assumed that subjects learned in units of reward, using a Rescorla-Wagner learning rule[14,64] which is applied at every second. After each second, the value of the environment is updated according to the following rule:

$$\rho_{t+1} = \rho_t + \alpha\delta_t. \tag{2}$$

Here $t$ indexes time in seconds. $\delta(t)$ is a prediction error, calculated as

$$\delta_t = r_t - \rho_t \tag{3}$$

$r_t$ is the reward obtained. This will either be 0 (for every second in which no reward is obtained, i.e. during search time, handling time, and timeouts from missed responses) or equal to $r_i$ (following receipt of a reward).

The learning rate $\alpha$ acts as a scaling parameter and governs how much participants change their estimate of the reward rate of the environment ($\rho$) from 1 s to the next. This estimate increases when $r$ is positive (i.e. when a reward is obtained) and decreases every second that elapses without a reward.

We implemented two versions of this reinforcement learning model. A Symmetric Model, with only a single $\alpha$ and a modified version, an Asymmetric Model, which had two $\alpha$: $\alpha^+$ and $\alpha^-$. In this second model, updates to $\rho$ (Eq. (2)) apply $\alpha^+$ if $r_t > 0$ and $\alpha^-$ if $r_t \leq 0$. This second model allows updates to occur differently according to whether a reward is received in the environment or not. This is close to identical to updating $\rho$ contingent on whether the prediction error, $\delta_t$, is positive or negative[65–67], as for the vast majority of trials (95%) in which $r_t > 0$, it was also the case that $\delta_t > 0$ while in all trials (100%) in which $r_t = 0$, it was also the case that $\delta_t < 0$. We refer to the mean difference in learning rates as the learning bias ($\alpha^+ - \alpha^-$). A positive learning bias ($\alpha^+ > \alpha^-$) indicates that participants adjust their estimates of the environments reward rate to a greater extent when a reward is

obtained compared to when rewards are absent. The converse is true when the learning bias is negative ($\alpha^+ < \alpha-$). If there is no learning bias ($\alpha^+ = \alpha-$) then this model is equivalent to the simpler Symmetric Model with a single $\alpha$.

To account for both the delay and the reward received in the final second of handling time (when participants received a reward), this was modeled as two separate updates to $\rho$; one update from the delay (in which $\delta_t = 0 - \rho_t$) followed by a second update from the reward received (in which $\delta_t = r_i - \rho_t$). Swapping the order of these updates or omitting the first update (from the delay) altogether did not alter the pattern of results.

The probability of choosing to accept an option is estimated using a softmax choice rule, implemented at the final (2nd) second of the encounter screen as follows:

$$P(\text{accept}) = \frac{1}{1 + \exp\big(\beta_0 - \beta_1(r_i - c_i)\big)}. \tag{4}$$

This formulation frames the decision to accept an option as a stochastic decision rule in which participants (noisily) choose between two actions (accept/ reject) according to the respective value of each of them. The temperature parameter $\beta_1$ governs participants' sensitivity to the difference between these two values while the bias term $\beta_0$ captures a participant's general tendency towards accepting/rejecting options (independent of the values of each action). Note that under this formulation, negative values for $\beta_0$ indicate a bias towards accepting options and positive values indicate a bias towards rejecting options.

At the beginning of the experiment, $\rho$ was initialized to the average (arithmetic) reward rate across the experiment[11] (although clearly not realistic as a process level model, this was included for simplicity to avoid estimation pathologies and special-case model features associated with initial conditions). In subsequent environments, the average reward rate carried over from the previous environment. In other words, there was no resetting when participants entered a new environment. Rather, they had to unlearn what they had learnt in the previous environment.

For each participant, we estimated the free parameters of the model by maximizing the likelihood of their sequence of choices, jointly with group-level distributions over the entire population using an Expectation Maximization (EM) procedure[68] implemented in the Julia language[69], version 0.7.0. Models were compared by first computing unbiased per subject marginal likelihoods via subject-level cross-validation and then comparing these likelihoods between models (Asymmetric versus Symmetric) using paired sample $t$-tests (two sided).

To formally test for differences in learning rates ($\alpha^+$, $\alpha-$) we estimated the covariance matrix over the group-level parameters using the Hessian of the model likelihood[70] and then used a contrast to compute the standard error on the difference $\alpha^+ - \alpha-$.

**Simulations**. To examine the qualitative fit of each model to the data we ran simulations for both the Symmetric Model and the Asymmetric Model. For each simulation ($n = 1000$), we ran a group of 40 virtual participants. For each virtual participant, we randomly assigned a set of parameters ($\beta_0$, $\beta_1$ and $\alpha$ in the case of the Symmetric Model; $\beta_0$, $\beta_1$, $\alpha^+$ and $\alpha-$ in the case of the Asymmetric Model) from the best-fit parameters generated by the computational model (fit to actual participants choices) and randomly assigned the order with which they encountered the environments (either rich to poor or poor to rich). Then, we simulated the learning process by which $\rho$ evolved as options were sequentially encountered and stochastically accepted/rejected. The learning process was exactly as described for the respective computational models. Crucially, $\rho$ was initialized exactly as for the model and allowed to carry over between blocks. We then calculated the difference in average acceptance rates for each virtual participant over the four options (LDLR, LDHR, HDHR, and HDLR) between environments (poor minus rich). We then calculated the average difference score over participants in each order condition. We then averaged these two sets of acceptance rates over each of the simulations.

To quantify the cost of the learning asymmetry under asymmetric and symmetric learning (see Supplementary Information) we ran another set of simulations for the Symmetric Model and the Asymmetric Model. For each simulation ($n = 500$), we once again ran a group of 40 virtual participants with half ($n = 20$) assigned to the RichPoor condition and the other half ($n = 20$) assigned to the PoorRich condition. Here we used the average learning rates ($\alpha$, $\alpha^+$, and $\alpha-$) and slope ($\beta_1$) from Experiment 1 and fixed the intercept to 0 ($\beta_0 = 0$) in order to isolate the pure cost of a learning bias (i.e. independent of differences between models in the general tendency to accept/reject options). We then calculated the average amount earnt for each of the 40 subjects in each model (excluding force trials) and compared earnings between symmetric and asymmetric learners using an independent sample $t$-test (two tailed).

**Participants and procedure Experiment 3**. A total of 59 participants were recruited for Experiment 3. Twenty-one of these were excluded (identical exclusion criteria as for Experiments 1 and 2) leaving a final sample of 38 participants (mean [standard deviation] age: 34.63 [8.58]; 12 females). The experiment was exactly as described as for Experiment 1 and Experiment 2 with one key difference. In this version there were three environments instead of two. A rich environment and a poor environment (exactly as for Experiments 1 and 2) as well as a third HDLR environment. The only options participants encountered in the HDLR

environment were the worst (HDLR) option. Participants had 10 min in each environment. The ordering was either Rich, HDLR, Poor or Poor, HDLR, Rich.

**Statistics and reproducibility**. Each of the three experiments reported were run once with an independent group of participants. Experiment 2 is a near replication of Experiment 1—the only difference being the duration of the two blocks (Experiment 1: 15 min, Experiment 2: 10 min).

**Reporting summary**. Further information on research design is available in the Nature Research Reporting Summary linked to this article.

## Data availability
All data are available at https://github.com/NeilGarrett/PreySelection. The source data underlying Figs. 2a–d, 3a–d, 4a, b, Supplementary Figs 1a–d, Supplementary Figs 2, 3, Supplementary Figs. 4a, b, Table 1 and Supplementary Table 1 are provided as a Source Data file. A reporting summary for this Article is available as a Supplementary Information file. Source data are provided with this paper.

## Code availability
All code is available at https://github.com/NeilGarrett/PreySelection.

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

## Acknowledgements

This research was supported by a US Army Research Office grant (W911NF-16-1- 0474) to N.D.D. and a Sir Henry Wellcome Postdoctoral Fellowship (209108/Z/17/Z) to N.G. We would like to thank Scott Grafton and Neil Dundon for helpful insight and discussions.

## Author contributions

N.G. and N.D.D. conceived the study, designed the study and wrote the manuscript. N.G. collected and analyzed the data.

## Competing interests

The authors declare no competing interests.
