## [Peer Review File · Nature Communications]

Reviewers' comments:

Reviewer #1 (Remarks to the Author):

This is a timely work on an interesting and increasingly important topic: to what extent does real agents' behavior accord with (and deviate from) optimal foraging theory? Here, Garrett and Daw use data from behavioral studies to show that a reinforcement learning model with distinct learning rates for positive and negative events successfully accounts for choice biases observed when agents transition between environments with different mean reward rates. They then replicate this finding in a second experiment.

All in all, I find the result clearly presented and convincing. Apart from the fact that it's a bit of a slim result, I have no major concerns. I think this adds to our understanding of sequential choice behavior in changing contexts and helps link foraging studies to accounts of learning.

Minor:

P4, bottom: "We first conducted two online experiments..." I can't find in the text how the experiments differed. ?

Line 324: "unlearn that they were no longer" – is this right?

Figure 3: "Asymmetry" instead of "asymmetric"? I think "asymmetric" is used at least a few times in the text.

Fig 3d: If you're testing the bars for significance, would be good to show uncertainty in them.

Reviewer #2 (Remarks to the Author):

This is an interesting and well designed behavioral and modeling study that suggests asymmetric updating of positive versus negative information in the context of a foraging-like behavior. Similar asymmetric updating has been observed in many other contexts, yet this study appears to be a meaningful, though incremental, advance in showing that this asymmetry extends a foraging style task. I do have some questions about the procedure implemented and the authors' interpretation that the results do in fact support asymmetric updating per se.

Can the authors provide more evidence to support the claim that it is learning rate that is asymmetric per se, or could the behavioral effects equally be explained by sticky priors for good or rich environments set up in the initial learning block. One could imagine that people generally expect experiments of this type to be rewarding and so participants are reluctant to adjust their belief that the environment is good after starting in a rich block. The model nicely produces the block order effect, but is this asymmetry seen in the first block alone, or does it only emerge after the block change? Given that

the crux of this paper centers on the claim of asymmetric updating, it seems important to distinguish between these accounts. If these cannot be dissociated in this task, the authors should include a new condition with multiple block changes or gradually changing richness that can disentangle these explanations.

Why aren't positive and negative PEs modeled to occur at the time a new alien is seen, indicating if the reward rate is higher or lower than expected? I may have missed something (see next point), but it seems that this is the key time to learn if the trial is better or worse than the average estimate if the values of the aliens are known. Or is this naturally captured by the modeling approach here that models negative and positive prediction errors with respect to time and reward, respectively?

Some details of the experimental design are unclear. Are subjects simultaneously learning the delays and rewards associated with each alien, or have these been learned previously or instructed to them? This may help explain my confusion above.

The primary new result presented in this study seems to be predicted naturally from previous demonstrations of asymmetric updating in non-foraging contexts (Sharot, Palmintieri). Is the take home from this study that this asymmetric updating is so widespread that it even extends to foraging-like behavior or is there something deeper at stake in this study?

Reviewer #3 (Remarks to the Author):

Review for "Biased belief updating and suboptimal choice in foraging decisions" by Garrett and Daw. The authors ran three experiments to investigate value update in the experimental context of foraging decision. Overall several experiments they found that the 'patch' value is learned in a biased manner: the learning rate for positive prediction errors is higher. The demonstration relies on both out of sample model comparison and model falsification (RichPoor vs. PoorRich analysis). All the results replicate across experiments and the statistics are very solid, with almost all the effects at $p < 0.001$. The manuscript is very well written and very well organized. Beyond methodological and statistical quality, the article is very important for the field, as it shows that this valence-induced learning bias, already showed in belief and reinforcement learning extends to the foraging contexts and may have important consequences in terms of suboptimal choices. I only have very few suggestions that I believe could improve the manuscript:

Optimality vs. suboptimality.

I apologize if I missed the information, but I think that it would be nice to actually quantify the cost of this learning bias against a benchmark of a normative model and/or a model without bias. I also believe that the discussion should be expanded a bit to include Cazé & van der Meer (Biological Cybernetics 2013). How do the optimality considerations proposed in the context of RL translate to the foraging context? I am asking because there is no strong evidence of this adaptive modulation in the literature

(see Gershman 2013; Chambon 2019). Maybe reason why this allegedly optimal modulation is not found is that the task they used is not ecological (i.e., not foraging) and the authors should comment on that.

Asymmetry vs. perseveration

A recent paper by Katahira (*Journal of Mathematical Psychology*; 2018) challenges the idea of learning bias on the ground that they could derive from mis-fitting choice perseveration. I have several issues with the choice perseveration interpretation (in most of the experimental contexts where the positivity bias has been shown). I think that it could be informative for the debate in the field, if the authors could compare a model w/ perseveration parameter to the asymmetric model and show in the Supplementary that it cannot explain the main RichPoor PoorRich effect. If for some reason the model cannot be correctly estimated in this context (because of an interference with the beta0 parameter?) the authors could at least mention how they rule out this alternative interpretation in the discussion.

Definition of context

I am not sure that the present results really stands in opposition of those reported by Palminteri et al. (2015), where the context is defined as an option pair. I think that what can be defined as a learning 'context' is somehow hard to translate from a task to another where the temporal structure and the instruction differ. On the other side, the RichPoor PoorRich results looks a lot like a 'contrast' effect and it would be interesting to discuss to what extent it could be accounted for by a relative value learning model.

Further probing

As the authors correctly point out (lines 472-3), within subject, direct comparison of the RichPoor, PoorRich trend may prove useful (especially if model simulations show distinctive not trivial patterns). As the experiments are performed online and the analytical pipelines are very clear the authors could actually go the extra mile and test this assumption.

The beta0 parameter:

The beta0 changes from positive to negative when moving from the Symmetric to Asymmetric model. Why is that? How does this relate to the arguments in lines 532-539?

Asymmetric update and depression

I would be more careful in considering biased learning rates or beliefs as a feature of depression, as there is no strong evidence for that (by Chase et al. *Psychological Medicine* DOI: <https://doi.org/10.1017/S0033291709990468>).

Just as basic check (it seems quite clear from the behaviour): have the authors formally tested that the 'no reset of values' assumption was justified in terms of fitting?

Language

I am not a native speaker, but I have the impression that “sluggish” and “sandwiched” are rather colloquial terms (but I may be wrong).

Reviewers' comments:

Reviewer #1:

This is a timely work on an interesting and increasingly important topic: to what extent does real agents' behavior accord with (and deviate from) optimal foraging theory? Here, Garrett and Daw use data from behavioral studies to show that a reinforcement learning model with distinct learning rates for positive and negative events successfully accounts for choice biases observed when agents transition between environments with different mean reward rates. They then replicate this finding in a second experiment.

All in all, I find the result clearly presented and convincing. Apart from the fact that it's a bit of a slim result, I have no major concerns. I think this adds to our understanding of sequential choice behavior in changing contexts and helps link foraging studies to accounts of learning.

Minor Comments:

P4, bottom: "We first conducted two online experiments..." I can't find in the text how the experiments differed?

- ◇ The two experiments were identical except for the duration of each block. Experiment 1 comprised 2 blocks each lasting 15 minutes. Experiment 2 comprised 2 blocks each

lasting 10 minutes. We have now added a sentence in the main text making this more explicit (p. 5).

Line 324: “unlearn that they were no longer” – is this right?

We apologize if this phrase was confusing. We intended to convey that participants had to update beliefs in the second environment that had been established in the first environment they encountered (poor for the PoorRich group, rich for the RichPoor group). We have rewritten the sentence with the phrase removed (p. 9).

Figure 3: “Asymmetry” instead of “asymmetric”? I think “asymmetric” is used at least a few times in the text.

◇ We now use *Asymmetric Model* and *Symmetric Model* throughout the text, tables, figures and figure captions.

Fig 3d: If you’re testing the bars for significance, would be good to show uncertainty in them.

◇ We thank the reviewer for this suggestion. The bars in Figures 3c and 3d previously showed *summed* Leave One Out Cross Validation Scores. We have now replaced these with bars showing *mean* Leave One Out Cross Validation Scores along with error bars (standard error of the mean). The Leave One Out Cross Validation Scores presented in Table 1 and Supplementary Table 1 have also been replaced to show means rather than summed scores accompanied by their respective standard errors.

Reviewer #2:

This is an interesting and well-designed behavioral and modeling study that suggests asymmetric updating of positive versus negative information in the context of a foraging-like behavior. Similar asymmetric updating has been observed in many other contexts, yet this study appears to be a meaningful, though incremental, advance in showing that this asymmetry extends a foraging style task. I do have some questions about the procedure implemented and the authors’ interpretation that the results do in fact support asymmetric updating per se.

Can the authors provide more evidence to support the claim that it is learning rate that is asymmetric per se, or could the behavioral effects equally be explained by sticky priors for good or rich environments set up in the initial learning block. One could imagine that people generally expect experiments of this type to be rewarding and so participants are reluctant to adjust their belief that the environment is good after starting in a rich block. The model nicely produces the block order effect, but is this asymmetry seen in the first block alone, or does it only emerge after the block change? Given that the crux of this paper centers on the claim of asymmetric updating, it seems important to distinguish between these accounts. If these cannot be dissociated in this task, the authors should include a new condition with multiple block changes or gradually changing richness that can disentangle these explanations.

◇ We thank the reviewer for this suggestion. We reran both the Symmetric Model and the Asymmetric Model on the first block alone. This again revealed that the Asymmetric Model was a better fit to the data than the Symmetric Model (Experiment 1: $t(39) = 5.56$, $p < 0.001$; Experiment 2: $t(37) = 6.01$, $p < 0.001$, paired sample ttests comparing Leave One Out Cross Validation Scores for each model) with $\alpha^+ > \alpha^-$ in the Asymmetric

Model (Experiment 1: $z = 5.64$, $p < 0.01$; Experiment 2: $z = 9.96$, $p < 0.01$). We include this extra analysis in the revised manuscript (p. 9).

Why aren't positive and negative PEs modeled to occur at the time a new alien is seen, indicating if the reward rate is higher or lower than expected? I may have missed something (see next point), but it seems that this is the key time to learn if the trial is better or worse than the average estimate if the values of the aliens are known. Or is this naturally captured by the modeling approach here that models negative and positive prediction errors with respect to time and reward, respectively?

- This question actually touches on a couple of different aspects of the model, which we have endeavored to clarify on revision: both *what* information drives learning, and *when*. On “what,” the key decision variable in the prey selection problem (Stephens and Krebs, 1986) is the overall long-run reward rate, given the subject's own choices. This is because this variable quantifies the opportunity cost, relative to reward expectancy under the status quo, of pursuing each new prospect.
- For this reason, the reward from an option which is merely encountered, but not accepted, does not update the rate. (This property is sometimes known as “independence of inclusion from encounter rate” in the foraging literature.) This is the reason that we only account for rewards that are actually obtained, not when they are offered.
- But as for “when,” the actual temporal patterning of the updates, we don't mean any strong claim here. We only assess choice behavior once per trial (and aren't, for instance, measuring neural prediction errors at particular events) so the actual updating according to the monetary rewards and time costs could occur at any point or points within the cycle with algebraically equivalent effect on choice; all we require is that the positive PEs (accounting for money) and the negative PEs (accounting for delay) be differentially weighted. We now clarify this in the revised manuscript (p. 9).

Some details of the experimental design are unclear. Are subjects simultaneously learning the delays and rewards associated with each alien, or have these been learned previously or instructed to them? This may help explain my confusion above.

- We apologize to the Reviewer for not being clearer on this point. The delays and rewards associated with each option were not previously learnt or instructed. Participants sequentially learnt these during the task itself following an accept decision when they then observed the time required to capture the accepted option and, following this, observed the reward collected. We now make this more explicit in the revised manuscript (p. 17).
- We note that the main two models presented in the paper (Symmetric Model and Asymmetric Model) assume for simplicity that participants accurately know the rewards and delays associated with each option (alien) from the start of the experiment. The rationale for this simplification is that there are only four option types, each of which is consistently and deterministically associated with a fixed reward and delay throughout the whole experiment; adequate sampling of these is further ensured with forced choice trials. All this means that learning of the option properties themselves is not likely to take very long (and has no reason to fluctuate once learned), so we focus our modeling on the more difficult, higher-level learning problem of learning the overall choice policy (i.e., the transient reward rate) from the stochastic encounters.
- However, we get the same pattern of results if we drop this assumption and instead incorporate an explicit model of the incremental learning of aliens' properties (as well as the reward rate via feedback during the task). We present the results of these additional models in the Supplementary Material.

The primary new result presented in this study seems be predicted naturally from previous demonstrations of asymmetric updating in non-foraging contexts (Sharot, Palmintieri). Is the take home from this study that this asymmetric updating is so widespread that it even extends to foraging-like behavior or is there something deeper at stake in this study?

- ◇ We agree with the Reviewer that the results are consistent with these previous demonstrations of asymmetric updating. However, we believe that it is surprising and important to have observed asymmetric updating in a foraging setting. Unlike beliefs about oneself (Sharot) or the potential payout of different bandits (Palmintieri), the foraging setting captures a key aspect of many real-world choices humans and other animals undertake – including hiring and job choice, mate selection, and indeed most realistic choices – for which all of the different options are not presented simultaneously via an exhaustive “menu” but instead must be sought out at least somewhat serially. As a result, choices are heavily reliant on subjective beliefs about how good are the options that might materialize in the future. The existence of asymmetries in how these beliefs are tracked and updated is therefore a mechanism by which a wide range of suboptimal decisions could potentially arise. (Thus, the deleterious effects of asymmetry are much more dramatic than they are, for instance, in bandit tasks where they tend to have similar effects across each option, and cancel out.) Indeed, it has long been argued in ethology that these types of decisions are under considerable selective pressure, yet also subject to substantial, unexplained biases. We have sought to highlight why we believe asymmetric learning in this setting is of important theoretical significance in the discussion of the revised manuscript (p. 14-15).

Reviewer #3:

Review for “Biased belief updating and suboptimal choice in foraging decisions” by Garrett and Daw. The authors ran three experiments to investigate value update in the experimental context of foraging decision. Overall several experiments they found that the ‘patch’ value is learned in a biased manner: the learning rate for positive prediction errors is higher. The demonstration relies on both out of sample model comparison and model falsification (RichPoor vs. PoorRich analysis). All the results replicate across experiments and the statistics are very solid, with almost all the effects at $p < 0.001$. The manuscript is very well written and very well organised. Beyond methodological and statistical quality, the article is very important for the field, as it shows that this valence-induced learning bias, already showed in belief and reinforcement learning extends to the foraging contexts and may have important consequences in terms of suboptimal choices. I only have very few suggestions that I believe could improve the manuscript.

Optimality vs. suboptimality.

I apologize if I missed the information, but I think that it would be nice to actually quantify the cost of this learning bias against a benchmark of a normative model and/or a model without bias.

- ◇ Following the Reviewer’s helpful suggestion, we quantified the cost of asymmetric learning against a model without bias by simulating earnings under asymmetric and symmetric learning (using the average learning rates from each model). This analysis revealed that symmetric learners earn 10% more than asymmetric learners over the course of the experiment ($t(78) = 10.03$, $p < 0.01$, independent sample ttest comparing earnings under symmetric learning versus asymmetric learning). We include this new analysis in the revised manuscript (Results p. 11 and Methods p. 21).

I also believe that the discussion should be expanded a bit to include Cazé & van der Meer (Biological Cybernetics 2013). How do the optimality considerations proposed in the context of RL translate to the foraging context? I am asking because there is no strong evidence of this adaptive modulation in the literature (see Gershman 2013; Chambon 2019). Maybe the reason why this allegedly optimal modulation is not found is that the task they used is not ecological (i.e., not foraging) and the authors should comment on that.

- ◇ Thanks for the interesting idea. The literature the Reviewer references (Cazé & Van der Meer) uses simulations to show that in bandit tasks, a positive learning asymmetry can be both advantageous and disadvantageous over the long run if payouts from each bandit are rare or common respectively. As a result, a learning mechanism that enables the balance of learning asymmetry to adjust from a positive to a negative bias according to the frequency of bandit payouts would be advantageous. However, to date, empirical work testing this has failed to find evidence for such an adaptive adjustment in bandit tasks (although there is evidence that biases in updating beliefs about oneself do attenuate under threat: see Garrett et al., 2018).
- ◇ However, our sense is that the direct extension of the Cazé & van der Meer logic to the foraging setting is not straightforward. It appears that the main reason asymmetric learning is advantageous for bandits is that the biases it introduces are similar across each option, so that when options are compared, it can enhance discriminability without biasing choice. This is unlikely to be the case in the foraging setting since the average reward serves a unique role as an aspiration level against which all other options are compared. Thus biases in the average bias the overall “choosiness” of the agent relative to *all* prospects, with potentially substantial effects. This is indeed, we think, why the bias we observe here is not optimal in the current task but costly (see our response to the point above).
- ◇ In all, then, we very much agree that these results highlight the importance of future theory, in the spirit of Cazé & Van der Meer’s, unpacking the impact of asymmetric learning in the foraging setting and clarifying the situations in which it might be adaptive or predicted to adjust. In the revised manuscript, we have included a paragraph in the discussion that addresses this point and cites the literature suggested (pp. 15-16).

Asymmetry vs. perseveration

A recent paper by Katahira (Journal of Mathematical Psychology; 2018) challenges the idea of learning bias on the ground that they could derive from mis-fitting choice perseveration. I have several issues with the choice perseveration interpretation (in most of the experimental contexts where the positivity bias has been shown). I think that it could be informative for the debate in the field, if the authors could compare a model w/ perseveration parameter to the asymmetric model and show in the Supplementary that it cannot explain the main RichPoor PoorRich effect. If for some reason the model cannot be correctly estimated in this context (because of an interference with the beta0 parameter?) the authors could at least mention how they rule out this alternative interpretation in the discussion.

- ◇ We thank the Reviewer for this suggestion. Our intuition is that the reason learning rate bias and choice perseveration are linked in bandit tasks does not apply to the foraging setting (for similar reasons to those discussed above) and we also share the reviewer’s intuition that locally perseverative choice cannot explain blockwise order effects. To test these intuitions formally, in the **Supplementary Material** we now include a Perseverance Model which includes a perseveration parameter (*13.stick*) as an additional free parameter.
- ◇ Comparing Leave One Out Cross Validation scores between this and the Asymmetric Model (via paired sample ttests) revealed that the Asymmetric Model again provided

a superior fit to choices compared to the Perseverance Model (Experiment 1: $t(39) = -7.84$, $p < 0.001$; Experiment 2: $t(37) = -7.79$, $p < 0.001$). There was also no significant improvement in scores between the Perseverance Model and the Symmetric Model (Experiment 1: $t(39) = -0.59$, $p = 0.87$; Experiment 2: $t(37) = -1.62$, $p = 0.11$). Simulations revealed that the perseverance model was also not able to qualitatively capture the order effect we observed (see below, green circles represent the pattern of choices generated by simulations from the Perseverance Model).

◇ The model and the results are included in the revised manuscript (**Supplementary Material and Supplementary Fig. 4**).

Definition of context

I am not sure that the present results really stand in opposition of those reported by Palminteri et al. (2015), where the context is defined as an option pair. I think that what can be defined as a learning 'context' is somehow hard to translate from a task to another where the temporal structure and the instruction differ. On the other side, the RichPoor PoorRich results looks a lot like a 'contrast' effect and it would be interesting to discuss to what extent it could be accounted for by a relative value learning model.

- ◇ We have deleted the sentence in the discussion suggesting our results stand in opposition to those of Palminteri et al. (2015), by which we didn't mean to imply any strong disagreement. Clearly both show some sharing across contexts (especially on initial encounter from one to the other in a block design like ours); and the ability to differentiate them (especially when repeatedly switching between them in an interleaved design like Palminteri's) can coexist.
- ◇ Regarding the other point, we aren't quite sure what the reviewer has in mind with contrast effects (e.g., how they explain rich/poor asymmetry). On the relationship between the models generally, we do assume that our opportunity cost term, and Palminteri's context-relativizing term, are likely the same thing, though it plays a somewhat different role in a bandit task (where it baseline-normalizes bandit-specific value updates) versus a foraging task (where it serves as a comparator for option acceptance). We also think the asymmetric learning aspect we focus on could be added to Palminteri's learning model, but isn't already explained by it unaltered. Also, for what it's worth, we experimented with models in which the comparison between option values and reward rates arises when learning the aliens' values, more like the Palminteri model, rather than in the choice rule, but did not have much luck. Such a model doesn't provide a natural explanation, for instance, for one-trial within-block

learning effects across alien types. In any case, we have added a bit of discussion concerning the possible relationship between these models on page 14.

Further probing

As the authors correctly point out (lines 472-3), within subject, direct comparison of the RichPoor, PoorRich trend may prove useful (especially if model simulations show distinctive not trivial patterns). As the experiments are performed online and the analytical pipelines are very clear the authors could actually go the extra mile and test this assumption.

- ◇ We agree that it would be preferable to formally test this assumption. Unfortunately, however, the authors involved are now employed by separate institutions (Princeton and Oxford) in separate countries, and the funding under which this program of research was run was unexpectedly frozen and is currently overspent. Therefore, despite some aspects of the study being straightforward, as the reviewer points out, it would be logistically challenging in several other respects to run the additional study being proposed at this stage. Accordingly, and given that this point is logically subsequent to most of the findings here, we have retained it as a caveat to the current findings, which we try to expose clearly, and as a proposal for subsequent research.

The beta0 parameter:

The beta0 changes from positive to negative when moving from the Symmetric to Asymmetric model. Why is that? How does this relate to the arguments in lines 532-539?

- ◇ β_0 captures any overall tendency to reject options (or accept them, when the parameter is negative) beyond the trial by trial pattern accounted for by the offer values r_L and costs c_L . Learning rate asymmetry in the Asymmetric model biases the opportunity cost $c_L = t_L p$ upward, which tends to increase rejection overall. Importantly, because this effect arises from an optimistically biased average reward p , it is not the same on every trial (but, for instance, there is proportionally extra rejection for options with larger t_L), so the model accounts for these patterned rejections via asymmetric learning rather than positive β_0 .
- ◇ By comparison, the symmetric model cannot capture this pattern of data because it cannot produce asymmetric updating of p , and thus the value-related terms in the softmax c_L underpredict rejection overall. It compensates for this, crudely, with an increased β_0 .
- ◇ As we now clarify (p. 15), the same relationship (that optimistic updating implies inflated opportunity cost and, on average, over-rejection), but applied the other way, underlies the point in the discussion the reviewer flags. This is that widespread reports of under-rejection (i.e. overstaying) in patch foraging tasks would, if it is due to asymmetric updating, imply pessimistic updating.

Asymmetric update and depression I would be more careful in considering biased learning rates or beliefs as a feature of depression, as there is no strong evidence for that (by Chase et al. Psychological Medicine DOI: <https://doi.org/10.1017/S0033291709990468>).

- ◇ We are now more cautious about this and cite the paper the reviewer suggests (p. 16).

Just as basic check (it seems quite clear from the behaviour): have the authors formally tested that the 'no reset of values' assumption was justified in terms of fitting?

- ◇ To be clear, we think this assumption is directly justified by the significant order effect, since a qualitative prediction of the model that resets value at the start of each block is clearly that behavior in a block is unaffected by whether it occurs first or last. We also think that this sort of categorical, parameter-free model falsification on the basis of qualitative predictions is actually a more powerful and informative test, when it can be applied, than one based on model-fitting (Palmenteri et al. 2017).
- ◇ Nevertheless, we ran a variant of the Asymmetric Model in which the reward rate reset between environments. Cross-validated marginal likelihoods were lower (better) for the no reset model compared to the reset model in both datasets, and this fit superiority was significant when both experiments were pooled ($F(1, 76) = 4.46, p < 0.05$). Again, in context, we don't think this is cause for concern; we think one reason this particular way of doing the comparison is statistically weaker is that the margin in favor of the no-reset model differs between subgroups (resetting turns out to make a greater difference for richpoor than poorrich participants, because the former's resistance to updating drives the order effect), which increases the variance of the across-subject test on model fit scores. But of course, it is precisely this difference between the two groups of participants (the between-subject order effect) that rejects the order invariance predicted by the no-resetting model.

Language

I am not a native speaker, but I have the impression that “sluggish” and “sandwiched” are rather colloquial terms (but I may be wrong).

- ◇ We have replaced these terms in the revised manuscript.

***REVIEWERS' COMMENTS:

Reviewer #1 (Remarks to the Author):

The authors have satisfactorily addressed all my concerns.

Reviewer #2 (Remarks to the Author):

The authors have responded to all my questions/concerns and I support publication.

Reviewer #3 (Remarks to the Author):

The authors did a great job addressing my concerns. I believe the article may be published without further revisions.